# Uncorrelated Geo-Text Inhibition Method Based on Voronoi K-Order and Spatial Correlations in Web Maps

**Yufeng He** [1,2,3], **Yehua Sheng** [1,2,3,*], **Yunqing Jing** [1,2,3], **Yue Yin** [1,2,3] and **Ahmad Hasnain** [1,2,3]

[1]  School of Geography, Nanjing Normal University, Nanjing 210023, China; 181301016@stu.njnu.edu.cn (Y.H.); 181301017@stu.njnu.edu.cn (Y.J.); 171301020@stu.njnu.edu.cn (Y.Y.); 171302143@stu.njnu.edu.cn (A.H.)
[2]  Key Laboratory of Virtual Geographic Environment, Ministry of Education, Nanjing Normal University, Nanjing 210023, China
[3]  Jiangsu Center for Collaborative Innovation in Geographical Information, Resource Development and Application, Nanjing 210023, China
*  Correspondence: shengyehua@njnu.edu.cn

**Abstract:** Unstructured geo-text annotations volunteered by users of web map services enrich the basic geographic data. However, irrelevant geo-texts can be added to the web map, and these geo-texts reduce utility to users. Therefore, this study proposes a method to detect uncorrelated geo-text annotations based on Voronoi k-order neighborhood partition and auto-correlation statistical models. On the basis of the geo-text classification and semantic vector transformation, a quantitative description method for spatial autocorrelation was established by the Voronoi weighting method of inverse vicinity distance. The Voronoi k-order neighborhood self-growth strategy was used to detect the minimum convergence neighborhood for spatial autocorrelation. The Pearson method was used to calculate the correlation degree of the geo-text in the convergence region and then deduce the type of geo-text to be filtered. Experimental results showed that for given geo-text types in the study region, the proposed method effectively calculated the correlation between new geo-texts and the convergence region, providing an effective suggestion for preventing uncorrelated geo-text from uploading to the web map environment.

**Keywords:** geo-text; spatial autocorrelation; Voronoi k-order; volunteered geographic information; semantic analysis; text auto-classification

---

## 1. Introduction

Geo-text annotations are a type of map annotation that can describe map elements using unstructured texts, unlike traditional annotations consisting of structured words or numbers. Geo-text has two indispensable components: unstructured text and a location. Unstructured text can be a sentence, paragraph, chapter, or article. Nowadays, many geo-text annotation services based on crowdsourcing models have recently been provided by web map platforms [1]. Geo-text data provided by volunteers are often applied to enrich basic geographic data [2–6]. The volunteers who upload these data on the map platforms also have opportunities to express thoughts, feelings, and comments via unstructured texts, images, and videos. To some extent, because of the location of the geo-text annotation, each user-contributed text could be with geographical information to reflect what the point's function is. Two important kinds of annotation, POI (point of interest) geo-text annotation and personal geo-text annotation, are now widespread in web map services. In detail, POI geo-text [7] referring to users' feedback on online goods, places, and services, such as the restaurant comments submitted by customers via Baidu or Google Maps, however personal geo-text [8,9] referring to

comments or descriptions that are attached to specific geo-positions and are shared by the volunteers to express their experiences, such as the personal tags provided by Google Earth [10] and the geo-text from Twitter presented in OpenStreetMap [11]. No matter which type geo text belong to (directly labeled or indirectly transformed) in a map application, as a type of semantic description of geographical location, geotext with coordinate information has the ability to describe the geographical environment and further help users describe an environment in greater detail environment than is possible though limited cartographic elements. Geo-text can therefore significantly improve spatial descriptions by supplementing structured text with abundant unstructured text.

Although geo-text annotations are abundant, they are also disorganized. At present, almost 600 million people in China alone are sharing data via convenient mobile maps, and the supervision and administration of geo-text services are inadequate for this level of use. Geo-texts that are randomly or maliciously added by users will greatly reduce the quality of the map service's data, and comprehensive data analyses should consider a geo-text's semantics and spatial competition, as follows, before permitting the geo-text to be uploaded to a map.

Geo-text semantics should be correlated with neighboring geo-texts [7]. The great number of geo-texts already available enables semantic autocorrelation within a region, and when a new geo-text is uncorrelated with existing semantic autocorrelations, the geo-texts should be filtered before uploading. For instance, a geo-text relevant to a farm should be filtered before being added to a map of a commercial region consisting of various shops and restaurants in a city.

Although extensive research has been conducted on how to obtain geo-text [12–15] and the use of geo-texts [16–20], few experiments have been conducted to inhibit uncorrelated geo-texts before adding them to web maps. Compared with illegal texts, geo-texts are more common and may add irrelevant annotations to web maps. In order to prevent the addition of irrelevant geo-texts, semantic information in geo-text annotations and spatial autocorrelations within neighborhoods must be investigated according to the following principles:

(1) The geo-text semantics are quantified by a text auto-classification method. Automatic text classification methods tend to be mature and form a unified process [21], and can, therefore, be used to establish a possibility type vector for the geo-text annotation [22,23].

(2) Spatial autocorrelations generated by nearby geo-text annotations should reject other unrelated newly submitted geo-texts. A statistical method that can address spatial proximity is key to identifying appropriate regions of spatial autocorrelation.

Spatial proximity can be assessed using either distance-based or topology-based measurements. Distance-based measurement methods include the range distance [24] and k-nearest neighbor methods [25] and these methods are suitable for modeling continuous data. However, the range distance method considers only the influence of distance, ignoring vicinity relations and thus regarding features of similar distance and different proximity as having the same influence, which does not meet this study's requirements. K-nearest neighbor model regards the nearest points as the neighbors, without considering the direction of data distribution. Topology-based methods consider vicinity by linking a common point to a common edge, as in the rook-queen-bishop distance method [26–29], Delaunay triangulation [30], and Voronoi diagrams [31]. The rook-queen-bishop distance method applies a 0–1 range to construct a weighted matrix. Delaunay triangulation measurements assign element centroids to determine adjacency; for instance, if each point or centroid is a triangle node, the nodes connected by the sides of the triangle are treated as adjacent nodes [30]. However, Delaunay triangulation cannot express spatial isolation and encounters difficulty when quantifying larger neighborhoods. Voronoi diagrams also assign point features or feature centers and establish boundaries between the separated objects, defining geometric adjacency between separated features based on boundary sharing [31]. This method can automatically adjust for the spatial differences among irregularly distributed data points and uneven densities and is therefore suitable for establishing a neighborhood for newly submitted geo-text annotations.

Spatial autocorrelation considers that an observation point is affected by other nearby observation points [27,32]. Classical spatial autocorrelation statistics include Moran's I [33,34], Geary's C [35] and Getis's G [36]. A positive spatial autocorrelation represents the clustering of similar values, and a negative spatial autocorrelation represents a discrete spatial distribution. Moran's I and Geary's C are global spatial autocorrelation statistics that estimate the degree of spatial autocorrelation of the overall dataset. Local Moran's I values provide estimates at the spatial unit level [37], and Getis's G calculates the degree of autocorrelation by comparing the neighborhood with the global average. In addition, many studies have also been conducted on the statistical significance of combinations of Moran's I spatial autocorrelations [38,39], and spatial autocorrelation methods are widely used in many fields [39–43].

The data quality in web maps has been reduced by the uploading of uncorrelated geo-text annotations, and filtering only illegal geo-texts is insufficient to correct the problem. Although geo-texts that support violence, illegal drugs, pornography, and other socially undesirable activities should clearly be filtered, these subjects have been addressed elsewhere [23,44,45], and will not, therefore, be discussed here. Therefore, this paper proposes an approach to improve the quality of geo-text annotations. First, geo-text annotation types are identified by a text auto-classification method, and the relationship between the existing semantic environment and a new geo-text can then be assessed by a correlated geo-text detection algorithm. A web map platform may then refuse to upload a geo-text, depending on the determined relationship.

The remainder of this paper is arranged as follows. Section 2 presents the geo-text classification method and filter algorithm, and Section 3 experimentally verifies the effectiveness of the proposed method. Section 4 concludes the experiment results discussion, and Section 5 includes the conclusion and identifies future research opportunities.

## 2. Uncorrelated Geo-Text Detection Method

Detection of semantically correlated geo-texts ought to seek semantic signals and spatial positions to (1) determine and categorize the geo-text's subject, usually by natural language processing and sophisticated semantic analysis techniques; (2) judge whether the geo-text is correlated with neighboring data at the semantic level, allowing upload based on spatial autocorrelation in a spatially aggregated region. According to Tobler's First Law of Geography, the closer the distance is, the more homogeneous and correlated the features tend to be. Spatial partitioning, distance measurements, and spatial autocorrelation analyses can be used to deduce the degree of correlation between a newly geo-text and its neighbors.

To develop a filtering method for uncorrelated volunteers' geo-texts on two-dimensional web maps, this section analyzes automatic geo-text classifications and then presents the proposed uncorrelated geo-text filter algorithm, which calculates the degree of correlation between a single geo-text and neighboring geo-texts. The overall method is shown in Figure 1.

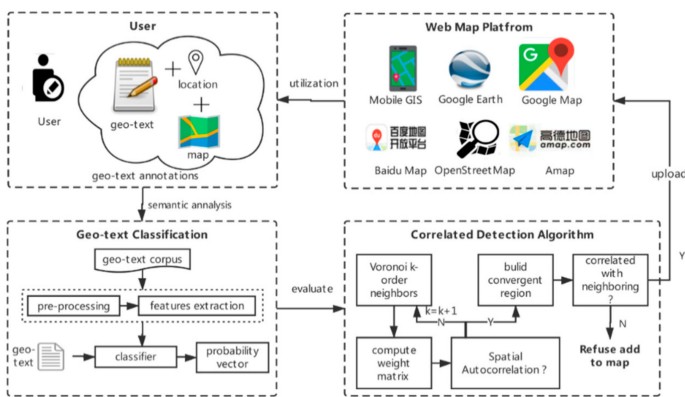

**Figure 1.** Overall flow of the proposed method.

*2.1. Automatic Classification for Geo-Text Annotations*

2.1.1. Geo-Texts Types

The geo-text annotation is a literal description of entities. Even if without structural attribute to indicates what type the entity belongs to, the functional type of geo-text by means of textual semantic analysis can be inferred. According to the Chinese National Language Commission Corpus and International Standard Industrial Classification of All Economic Activities, text could be divided into different types. It also works to geo-text annotations.

A detailed geo-entity classification system provided by Baidu Map brings convenience to map users and meets the requirements for the classification for geographical text annotation types. The first level of urban geo-texts included restaurants, hotels, life service, tourist spots, entertainment, educational institutions, cultural media organizations, hospitals, houses, government agencies, and so on. The second level was more detailed, distinguishing among traditional restaurants, foreign restaurants, fast food restaurants, cake shops, cafés, tea houses, and bars in the restaurant category; among apartments, dachas, high-rise buildings, and multistoried buildings in the housing category; and among institutions of higher learning, middle schools, primary schools, and research institutions in the school category. The more detailed catalog information we can see from the website ('http://lbsyun.baidu.com/index.php?%20title=lbscloud/poitags').

2.1.2. Automatic Classification Method

Conventional geographic annotation (*ga*) can be formalized as follows:

$$ga = \{(x, y), term\}c \tag{1}$$

where (*x*, *y*) refers to the location of *ga* and *term* is the content of the geographically structured annotation (a term or number). This annotation can be extended to include the geo-text annotation (*gta*) as follows:

$$gta = \{(x, y), text\} \tag{2}$$

where *text* may be an unstructured word, sentence, paragraph, or even article. The ordered vector of each word represents the whole semantic meaning of the sentence or text. Through the word segmentation process, unstructured text can be described as a structured word sequence vector [46]:

$$gta = \{(x, y), (term_1, term_2, \dots, term_n)\}, n \in \mathbf{N}. \tag{3}$$

In this equation, a geo-text annotation vector consisting of the statistical frequency of different *terms* is the eigenvector of *text*. Machine learning methods typically used for text auto-classification [47,48] can identify the type of geo-text annotation. In general, the steps for text auto-classification include: pre-processing by eliminating stop-words [49], stemming [50], feature selection using various statistical or semantic approaches [51,52], and modeling using appropriate machine learning techniques such as naïve Bayes [53], neural network [54], support vector machines [55], and hybrid techniques [56]. Next, a term frequency-inverse document frequency (TF-IDF) eigenvector is selected for the geo-text annotation. This is a commonly used weighting technique for information retrieval and data mining, whereby the importance of a word is evaluated according to its occurrence frequency in documents and a related corpus [57]. Finally, a multinomial naïve classifier is applied. The naïve Bayesian classifier is based on the assumption of conditional independence among attributes [22,58]. As the naïve Bayesian approach is purely statistical, its implementation is straightforward and requires less learning time, making it suitable for classifications of short text segments [22]. Figure 2 summarizes the process of geo-text annotation classification.

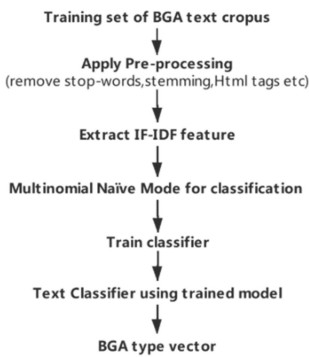

**Figure 2.** Flow of geo-text annotation classification.

Based on the text features established by the evaluation, a multinomial naïve classifier is adopted. Therefore, vector $V_k$ of the geo-text annotation belonging to class $k$ is described as:

$$V_k = \{(term_1, term_2, \ldots, term_n)\} = \{(w_{k1}, w_{k2}, \ldots, w_{kn})\} \tag{4}$$

where $y$ is the class of $gta$, $n$ is the number of terms, and $w_{ki}$ is the probability that $i$ belongs to class $k$, written as $P(x_i|y)$. The relationship between class and the related eigenvectors $(x_1, x_2, \ldots, x_n)$ is as follows:

$$P(y|x_1,\ldots,x_n) = \frac{P(y)P(x_1,\ldots,x_n|y)}{P(x_1,\ldots,x_n)} \tag{5}$$

Due to the independence imposed by the naïve conditions,

$$P(x_i|y, x_1,\ldots,x_{i-1}, x_{i+1},\ldots,x_n) = P(x_i|y) \tag{6}$$

$$P(y|x_1,\ldots,x_n) = \frac{P(y)\prod_{i=1}^n P(x_i|y)}{P(x_1,\ldots,x_n)} \tag{7}$$

$P(x_1,\ldots,x_n)$ is constant, and therefore the classification rules are as follows:

$$P(y|x_1,\ldots,x_n) \propto P(y)\prod_{i=1}^n P(x_i|y) \tag{8}$$

$$\hat{y} = \underset{y}{\operatorname{argmax}} P(y)\prod_{i=1}^n P(x_i|y) \tag{9}$$

where $P(y)$ is the frequency of $y$ in the training set. A maximum posterior estimation is used to estimate $P(x_i|y)$. Knowing that $n$ is the number of $y$ classes and $y_k$ is the class $k$, we can calculate the probability of word vectors in $k$ class $P(y_k)$:

$$P(y_k) = P(y)\prod_{i=1}^n P(x_i|y_k). \tag{10}$$

We then calculate the probability value of each class, getting the probability vector $vc$ of each class after normalization:

$$vc = \left[\frac{P(y_1)}{\sqrt{P(y_1)^2 + \ldots + P(y_1)^2}}, \ldots, \frac{P(y_k)}{\sqrt{P(y_1)^2 + \ldots + P(y_1)^2}}, \ldots, \frac{P(y_n)}{\sqrt{P(y_1)^2 + \ldots + P(y_1)^2}}\right]. \tag{11}$$

The type $\hat{y}$ of word vector $(term_1, term_2, \ldots, term_n)$ is computed as follows:

$$\hat{y} = \underset{y_k}{\operatorname{argmax}} \left\{\frac{P(y_1)}{\sqrt{P(y_1)^2 + \ldots + P(y_1)^2}}, \ldots, \frac{P(y_k)}{\sqrt{P(y_1)^2 + \ldots + P(y_1)^2}}, \ldots, \frac{P(y_n)}{\sqrt{P(y_1)^2 + \ldots + P(y_1)^2}}\right\} \tag{12}$$

Geo-texts are considered to belong to the type $\hat{y}$ for which the vector component value is largest. When $\hat{y}$ is considerably larger than the other vector component values, the classification is more accurate. For ease of calculation, *vc* was magnified 100 times.

## 2.2. Correlated Geo-Text Detection Algorithm (CGD)

On the basis of the derived geo-text probability vector, we propose a correlated geo-text filter algorithm based on autocorrelation and Voronoi k-order, namely the correlated geo-text detection (CGD) algorithm. The algorithm proceeds in the following four steps: (1) Use Voronoi k-order adjacent methods to establish a self-growing nested neighborhood for geo-text positioning; (2) select a spatial weighting method suitable for the Voronoi k-order neighborhood and build a spatial weighting matrix in the neighborhood to support a quantitative spatial correlation calculation; (3) detect the minimum neighborhood of spatial convergence based on a global Moran's I at a statistically significant level; (4) calculate the degree of correlation between a geo-text annotation and its neighborhood using the Pearson correlation coefficient method. The flow of this algorithm is shown in Figure 3.

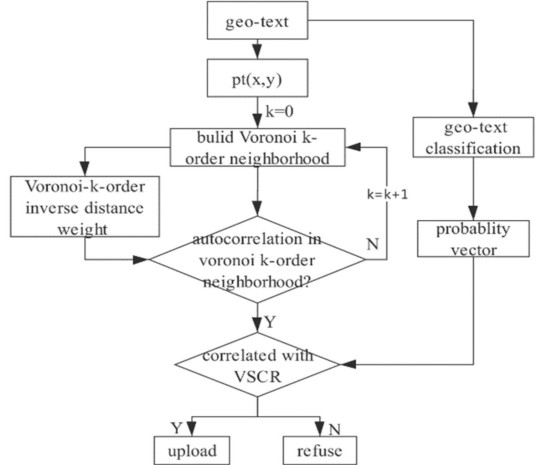

**Figure 3.** Flow of proposed Correlated Geo-text Detection Algorithm (CGD).

### 2.2.1. Voronoi k-Order Neighborhood Partition

There are two types of methods for evaluating spatial proximity: distance relations and topological relations. Voronoi k-order methods are based on topological relations and are more suitable for describing local data [31,57]. The Voronoi k-order neighborhood partition method is used to establish a nested neighborhood structure centered on the geo-text annotation and gradually expanding outward. This structure is beneficial to find the minimum convergence neighborhood around the geo-text. The Voronoi k-order vicinity is defined as follows. For the spatial dataset: $\mathbf{O} = \{o_1, \ldots, o_n\} \subset \mathbf{R}^2$, $1 < n < \infty$, $\forall o_i, o_j(i \neq j)$. If $o_i$ passes the minimum Voronoi adjacent steps $k(k \in \mathbf{N})$ to reach the target $o_j$, $o_i$ and $o_j$ are Voronoi k-order adjacent to each other.

In a finite space $\mathbf{R}^2$, the spatial point dataset $\mathbf{P} = \{p_1, p_2, \ldots, p_i, \ldots, p_j, \ldots, p_n\}$, and $v(p_i)$ and $v(p_j)$ are Voronoi regions of $p_1$ and $p_2$, respectively. The Voronoi k-order adjacent from $p_m$ is then described as

$$N_k(P_m) = \left\{ v(p_i) \cap v(p_j) \neq \varnothing, p_j \in N_{k-1}(P_m), k > 0 \right\}$$
$$N_0(p_m) = p_m \tag{13}$$

Therefore, the Voronoi k-order adjacent from $p_m$ can also be expressed as:

$$N_k(P_m) = \{P_i | vd(P_i, P_m) = k, k > 0\}. \tag{14}$$

As shown in Figure 4, assuming that a geo-text annotation is added at point 0, the Voronoi 0-order adjacent features (green), Voronoi 1st-order adjacent features (pink), Voronoi 2nd-order adjacent features

(orange), and Voronoi 3rd-order adjacent features (blue) are established according to Equation (14), namely: $v$ (0), $v$ (1), $v$ (2), and $v$ (3), respectively.

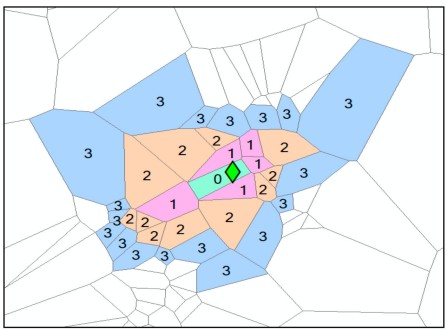

**Figure 4.** Voronoi k-order adjacent model.

The Voronoi k-order neighborhood (written as $vn(k)$) of a geo-text annotation includes all of the features from Voronoi 0-order to the Voronoi $k$-order, following the equation: $vn(k) = v(0) + v(1) + \ldots + v(k)$. Therefore, as shown in Figure 5, the regions of the Voronoi 1st-, 2nd-, and 3rd-order neighborhoods are $vn(1)$, $vn$ (2), and $vn$ (3) respectively.

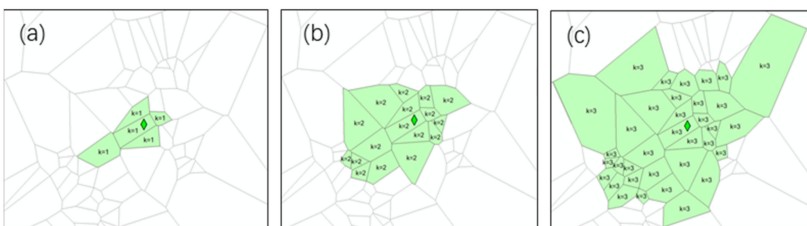

**Figure 5.** Voronoi k-order neighborhoods: (**a**) $vn(1)$, (**b**), $vn$(2), and (**c**) and $vn$(3).

### 2.2.2. Weight Matrix Based on Voronoi k-Order Neighborhood

Due to the contributions of geo-texts at different positions can be inconsistent, a weight matrix is used to consider all the weights in the neighborhood. For this purpose, a Voronoi k-order inverse distance weighting method that combines distance measurements with spatial vicinity is proposed.

The evaluation region is $vn(k)$, the range distance between $p_i(x_i, y_i)$ and $p_j(x_j, y_j)$ is $d(p_j, p_i)$, and the Voronoi vicinity distance is $vd(p_j, p_i)$. As shown in Figure 6, the range distance $d(p_j, p_i)$ is equal to 218.53 m and the Voronoi vicinity distance $vd(p_j, p_i)$ is equal to 3. The weight value $w_{ij}$ equation is:

$$w_{ij} = \begin{cases} \dfrac{1}{vd(p_i,p_j) \times d(p_i,p_j)}, & vd\left(p_i, p_j\right) < \tau_k \\ 0 \quad, & vd\left(p_i, p_j\right) \geq \tau_k \end{cases} . \tag{15}$$

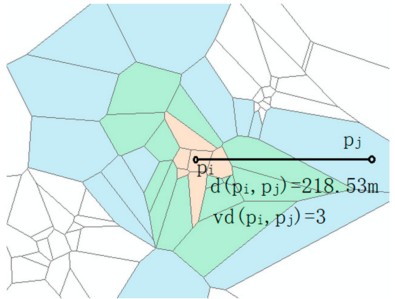

**Figure 6.** Voronoi k-order inverse distance weighting.

With $\tau_k$ as the threshold, when a Voronoi vicinity distance exceeds the threshold, the geo-text is considered too far away to influence a new geo-text annotation and therefore $w_{i,j}$ is set to zero. All weights relative to the target point are normalized in $vn(k)$, $\sum_j w_{i,j} = 1$. In the Voronoi k-order neighborhood, the weight matrix is constructed as $W$:

$$W = \begin{pmatrix} w_{11} & \cdots & w_{1n} \\ \vdots & w_{ij} & \vdots \\ w_{m1} & \cdots & w_{mn} \end{pmatrix}. \tag{16}$$

The anti-distance weighting of the Voronoi-k-order is between the anti-Euclidean distance and the anti-Voronoi k vicinity distance, which is more in line with human cognition.

### 2.2.3. Detecting the Minimum Voronoi-k-Order Semantic Convergence Region of A Point

Moran's I statistics are indicators that measure global spatial autocorrelation [36]. After establishing the neighborhood and spatial weight matrix of the geo-texts, the global Moran's I calculation formula [47] is as follows:

$$I = \frac{n}{S_0} \frac{\sum_{i=1}^{n} \sum_{j=1}^{n} w_{ij} z_i z_j}{\sum_{i=1}^{n} z_i^2} \quad S_0 = \sum_{i=1}^{n} \sum_{j=1}^{n} w_{ij} \tag{17}$$

where $x_i$ refers to the value of the probability vector for the geo-text in the neighborhood, in the paper, $x_i$ is a vector of $vc$ shown in Equation (11); $z_i$ is the difference between $x_i$ and its average $\overline{X}$; $w_{ij}$ refers to the Voronoi k-order anti-distance weight of $x_i$ and $x_j$, from Equation (15); $N$ is the number of features; $S_0$ is the sum of weights in the neighborhood. Moran's I can interpret spatial dispersion and aggregation only under the statistical significance. The $z_I$ score is calculated as follows:

$$z_I = \frac{I - E[I]}{\sqrt{V[I]}} \quad E[I] = -1/(n-1) \quad V[I] = E[I^2] - E[I]^2. \tag{18}$$

Under the condition that $p$ is at a statistically significant level, if I > 0 and z > 0, the spatial distribution of high or low values in the dataset has a higher clustering degree than expected. If z < 0, the spatial distribution of high or low values in the dataset is more discrete in space than expected [59]. If I = 0 and z = 0, the space is randomly distributed. Combining geo-text neighborhoods partitioned by Voronoi vicinity and the condition (I > 0 and z > 0) of spatial autocorrelation, a semantic autocorrelation region in the neighborhood can be established. Under the condition of rejecting the non-null hypothesis, the minimum Voronoi k-order neighborhood that delineates the space of the geo-text semantic autocorrelation, namely the Voronoi-k-order semantic convergence region (*VSCR*), is:

$$\begin{array}{c} VSCR = \left\{ bta \mid vd\left(bta_i,\ bta_j\right) = k \right\} \\ k \le argmin(I_i(VSCR) > 0 \bigcap Z(I_i) > 0 \bigcap p < 0.1) \end{array}. \tag{19}$$

Minimum VSCR helps to deduce what semantic type the new geo-text location is.

Moran's I cannot be calculated when the components of a geo-text vector in the neighborhood are nearly equal. For example, many comments about shops appear in the mall, whereas comments about companies appear in industrial parks, and flower descriptions in gardens. Other types of geo-texts will be excluded if a region consists of the same type of points, according to the restraint of spatial autocorrelation. Therefore, we must take action to detect the variant phenomenon.

There are two types of variation that should be operated before searching *VSCR* for geo-texts. First, in high-value aggregation distribution, when detecting the vector component from the Voronoi 1st-order to the Voronoi k-order (k ≥ 3), if all of the given geo-text component types are greater than 70% (an empirical value), other types of geo-texts are filtered according to the autocorrelation rule. In the second type of variation, low-value aggregation distribution, when detecting a vector component

region from the Voronoi 1st-order to the Voronoi *k*-order (k ≥ 3), if all the components of a given type are less than 20% (an empirical value), a low-value aggregation distribution is formed, and this given type of geo-text will be rejected from the map.

2.2.4. Similarity Analysis of Geo-Text Annotation in VSCR

As shown in Figure 7, there are two types of labeling situations: either the new geo-text annotation is placed in a neighborhood containing similar types (a single-type neighborhood) or it is placed in a neighborhood containing different types (a compound-type neighborhood). Single-type neighborhood annotation is a special form of compound-type annotation.

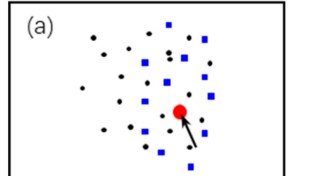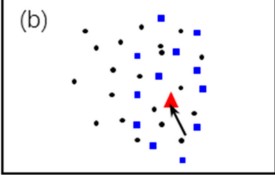

**Figure 7.** Geo-text annotations in (**a**) a single-type neighborhood and (**b**) a compound-type neighborhood.

When the new geo-text annotation is placed in a single-type neighborhood, the mean and variance of variables in the neighborhood will not change significantly, and the situation conforms to the First Law of Geography. In contrast, for compound-type neighborhoods, the new geo-text will break the original distribution, and may therefore not be allowed at this position, being an uncorrelated geo-text. For example, in Figure 7, blue squares represent restaurants and black circles represent department stores and red triangles represent farms, it is very easy to allow that the new geo-text of department store add to this point by the reason of according with the same single-type neighborhood in *VSCR*, and also prohibited to add farm geotext because of the farm's nonexistent of semantic environment in its *VSCR*.

A probability histogram transformed from the geo-text probability vector *vc*, shown in Equation (12), contributes to detecting correlations between new geo-texts and the potentially relevant neighborhoods. The probability histogram of $p_m$ in region *vn(k)* is *Hist*($p_m$); the geo-texts dataset **P** = {$p_1, \ldots, p_i, \ldots, p_j, \ldots, p_n$} in *vn(k)*, and its histogram set is: $\textbf{Hist}_p$ = (*Hist*$_1$, … , *Hist*$_i$, … , *Hist*$_j$, … , *Hist*$_P$), where *k* refers to the histogram bin sequence and *Hist*$_i$ is equal to *vc* of geotext $p_i$, and the correlation degree between the new geo-text and its *VSCR* is calculated by the Pearson correlation coefficient. Note, because *vc* is the normalized input-params, the Pearson correlation coefficient can be used to judge the similarity between two geotext annotations, just as it can be used to judge the similarity between two image histograms.

$$corr(p_m, VSCR) \;=\; \mathrm{argmax}_{i=1}^{n} \frac{\sum_k (Hist_m(k) - \overline{Hist_m})(Hist_i(k) - \overline{Hist_m})}{\sum_k \left[ (Hist_m(k) - \overline{Hist_m})^2 (Hist_i(k) - \overline{Hist_m})^2 \right]^{1/2}} \tag{20}$$

Within $\overline{Hist_i} \;=\; \frac{1}{N} \sum_{j=1}^{N} Hist_i(k)$, N is the number of histogram bins.

$$Sim(p_m, N_k(p_m)) \;=\; \begin{cases} false, & Corr(p_m, GTASC) < \tau \\ true, & Corr(p_m, GTASC) \geq \tau \end{cases}. \tag{21}$$

When the correlation between $p_m$ and its *VSCR* is higher than the threshold $\tau$, the geo-text is correlated with the neighborhood. Conversely, when the $p_m$ and the *VSCR* are mutually uncorrelated, the new geo-text is refused from the map at this $p_m$.

There are four category features, for instance, restaurant, school, house and hotels. Figure 8a shows the *hist*($p_m$) is (78,14,6,2), which we convert a geotext to probability vector according to Equation (11), and Figure 8b shows the histogram value of any geo-text $p_i$ in the *VSCR* is (83,9,3,5). The Pearson correlation coefficient between $p_m$ and $p_i$ is 99.52%, when the threshold $\tau$ is 0.9, the geo-text is correlated with the neighborhood and geotext of $p_m$ can be added.

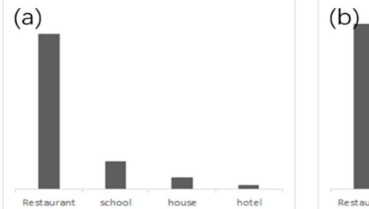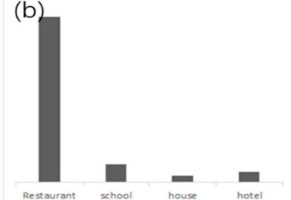

**Figure 8.** Histogram correlation analysis between (**a**) *Hist*($p_m$) and (**b**) and *Hist*($p_i$).

Instead of using absolute type (select biggest **vc** component as classification type) to search similar annotations, the Person correlation coefficient method can distinguish some extreme cases. For example, the two annotations with classification probability **vc** (90, 5, 5, 0) and **vc** (49, 48, 1, 2) respectively are quite different. The first one was clearly classified, while the second was ambiguous. It is wrong when they were classified as the same type by means of an absolute type comparison method. By using the correlation coefficient method, their similarity is only 59.6%, which is in line with our actual cognition.

## 3. Experiment Validation

The validation experiment tested the proposed auto classification and CGD algorithm on POI geo-texts taken from the Baidu Map platform. We applied POI geo-texts because these geo-texts, which are provided by volunteers, were very dense and abundant. Any new geo-text can build the *VSCR* with sufficient geo-texts in the neighborhood to detect correlations. The geo-text classification method and CGD algorithm were created in Python 3.6, and the jieba, scikit-learn, and PySAL packages were used in this experiment.

The experimental area was located in Xianlin, a university town in the Qixia district of Nanjing, composed mainly of schools, houses, and restaurants. We collected 631 records from BaiduMap and the Popular Comments App in the experimental area and transformed the geo-text points to the WGS 84 projection coordinate system. These transformed geo-text points were then drawn on Baidu Map (Figure 9).

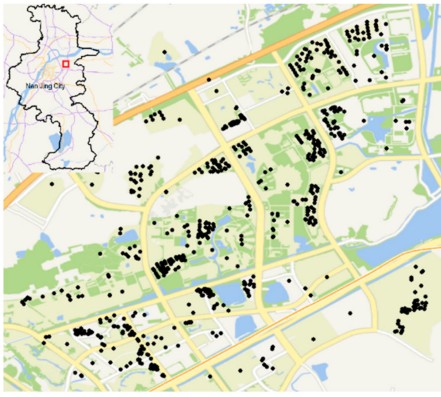

**Figure 9.** Geo-text points on Baidu Map.

### 3.1. Auto Classification of Geo-Texts

According to the overall distribution of the types of geo-text annotation in the study area, the groups of types were selected for the experiment of anomaly geo-text annotation detection. This experiment geo-text group included restaurants, houses, schools. As Chinese restaurants were included as restaurants, the group that consisted of Chinese restaurants, restaurants, schools, houses was not available. After knowing the types of POIs in the research area, the next step is sample training.

### 3.1.1. Train Classifier

The corpus was established by crawling 1000 Chinese geo-text instances for each geo-text type which was chosen from the Baidu Map API, and then geo-texts points were transformed to a WGS 84 projection coordinate system. The sample consisted of 1000 house texts, 1000 restaurant texts and 1000 school texts were applied to train in the multinomial naive Bayes classifier. Sample and test data were extracted at a ratio of 4:1. The shortest geo-text was 30 bytes, and the longest was 1372 bytes, with a mean length of 205 bytes. After word segmentation and stop-word dropping, there were 40958 columns and 2756 rows in TF-IDF features vectors. Six hundred geo-text annotations were used to evaluate the accuracy of the multinomial naive Bayes classifier. According to the classification results, the classification accuracy was 97.78%, the recall rate was 97.79%, and the F-score was 96.83%. Therefore, the corpus and classifier used in this paper were considered to achieve effective classification for geo-text.

### 3.1.2. Geo-Text Classification Process

Three geo-texts A, B, and C, which we crawled randomly from house POIs, school POIs and restaurant POIs respectively on the Baidu Map, were applied to verify the process of geo-text classification. Chinese geo-text A, B and C shown in Figure 10. Geotext A was translated into English as follows: "It is a very beautiful village in Xianlin university town, located in a prosperous area. Now the average price of housing is more than 30,000 yuan. The transportation there is convenient. Primary schools, kindergartens and universities nearby village, so the cultural atmosphere there is very good." The vector $vc$ was (78.419, 21.581, 0), and the conclusion was that the type of point A was a house. Point B was described thus: "The school consists of eight departments: geography, tourism, land management, geographic information system, remote sensing, environmental science and engineering, and marine science. Geography was rated as double first-class discipline, cartography and geographic information system as the national key discipline and the key discipline of Jiangsu province, and physical geography as the key discipline of Jiangsu province." The $vc$ was (0, 90.302, 9.608). Therefore, B was related to the school type. Chinese geo-text C was translated as: "I often eat here, delicious also dish quantity, very affordable. The shopkeeper's service is very warm. I recommend the sauce bone, pot wrapped meat, stir-fried chicken and northeast sauce bone. All of them are very delicious, and the price is moderate." The $vc$ was (15.32, 1.29, 83.39), and C was considered a restaurant type. The elected biggest $vc$ component as a classification type, Geo-texts A, B, and C were correctly classified, and their classification probability vector was computed.

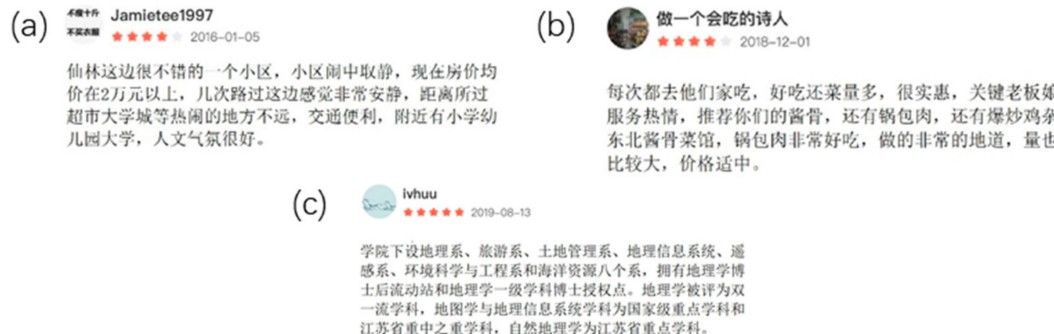

**Figure 10.** Chinese geo-text A (**a**), Chinese geo-text B (**b**) and Chinese geo-text C(**c**).

All the geo-texts in the experimental area were similarly classified and converted to vectors via the text auto-classification method, and school, house, and restaurant geo-texts were plotted in red, green and blue, respectively, as shown in Figure 11.

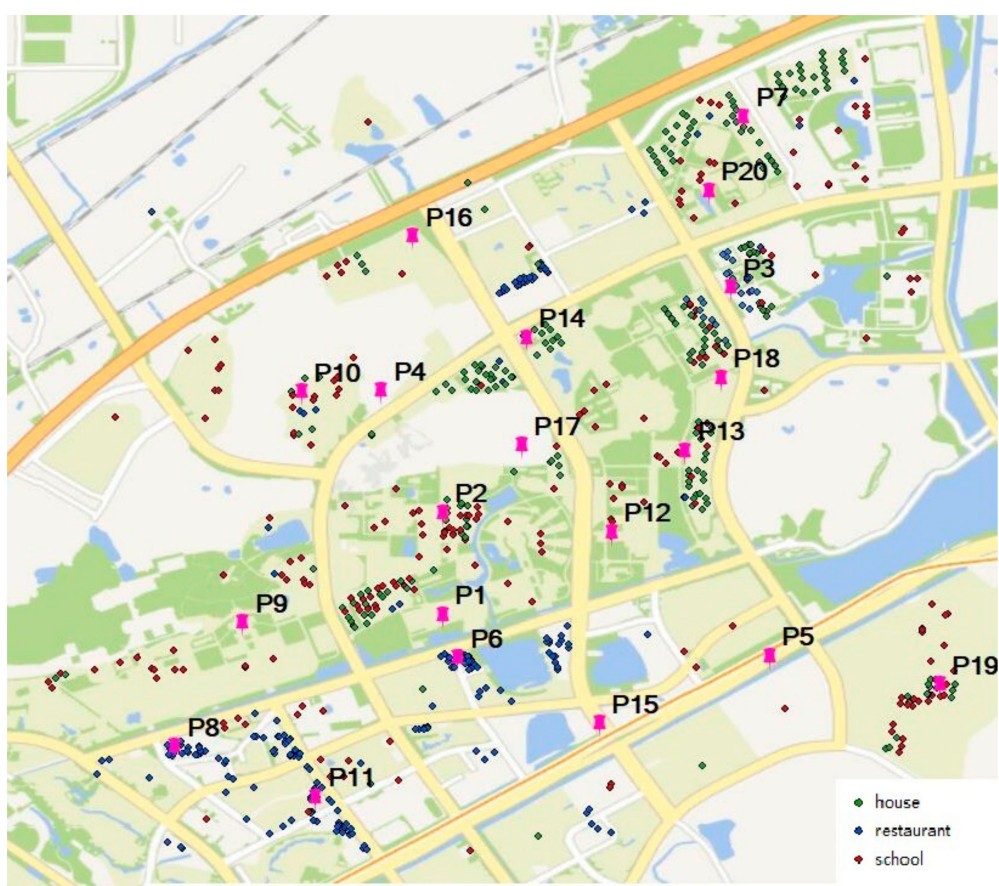

**Figure 11.** Geo-text classification results.

## 3.2. Correlated Geo-Text Detection (CGD) Algorithm Experiment

### 3.2.1. CGD Algorithm Process

In order to validate the CGD algorithm proposed in this paper, the house geo-text annotation A was placed in P1. First, we built a Voronoi k-order nesting neighborhood for p1: $vn(1)$, $vn(2)$, $vn(3)$, $vn(4)$, $vn(5)$, $vn(6)$, shown in Figure 12. According to the anti-Voronoi distance weight matrix, the values of $I$, $p$, and $z$ from the Voronoi 1st-order to the Voronoi k-order neighborhoods were calculated respectively, as shown in Table 1. The values of $I$, $p$, and z were then compared with the convergence conditions ($I > 0$, $z > 0$, and $p < 0.1$), and $nv(k)$, where $k$ is the minimum value for reaching the requirement, was

selected as the *VSCR*. For the P1 point, the *VSCR* was clearly *vn*(2). The correlation degree between type vector **vc** (78.419, 21.581, 0) and the other geo-texts in *vn*(2) was then calculated via the Pearson correlation coefficient method. There were two geo-texts similar to the *VSCR*, which consisted of 21 geo-texts. Therefore, A was detected as correlated with the neighborhood and allowed to upload to the web map services. Figure 13 shows Moran's I in different Voronoi k-order nesting neighborhoods.

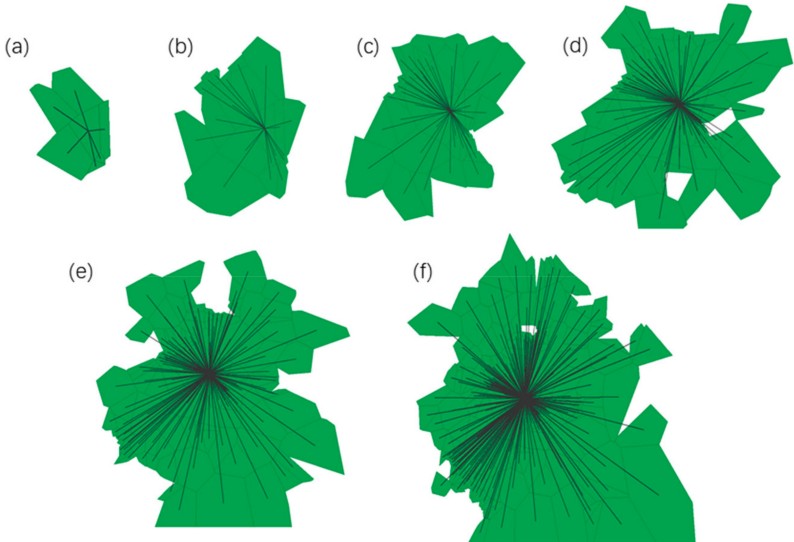

**Figure 12.** k-order neighborhood of geo-text A in (**a**) vn(1), (**b**) vn(2), (**c**) vn(3), (**d**) vn(4), (**e**) vn(5), and (**f**) *vn*(6) of P1.

**Table 1.** Moran's I of A in k-order of P1.

| k | 1 | 2 | 3 | 4 | 5 | 6 |
|---|---|---|---|---|---|---|
| Count | 7 | 21 | 47 | 88 | 145 | 225 |
| I | −0.131 | 0.227 | 0.201 | 0.332 | 0.358 | 0.425 |
| p | 0.181 | 0.001 | 0.014 | 0.001 | 0.001 | 0.001 |
| z | 0.353 | 2.667 | 3.796 | 8.071 | 10.116 | 13.909 |

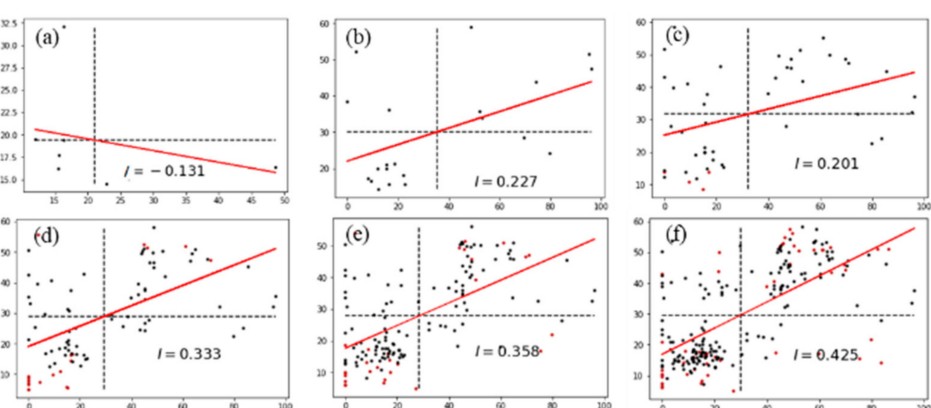

**Figure 13.** Moran scatterplot of geo-text A in (**a**) *vn*(1), (**b**) *vn*(2), (**c**) *vn*(3), (**d**) *vn*(4), (**e**) *vn*(5), and (**f**) *vn*(6) of P1.

The results of geo-texts B and C were similarly calculated, and both of them were spatially autocorrelated in *vn*(2). In other words, their *VSCR* was *vn*(2), in which seven of the 21 geo-texts correlated with B, and 12 of the 21 geo-texts correlated with C. Therefore, B and C were detected as correlated with the neighborhood and allowed to upload to the web map. Figure 14 shows the trends of I, *p*, and *z* values for A, B, and C by increasing Voronoi k-order nesting neighborhoods.

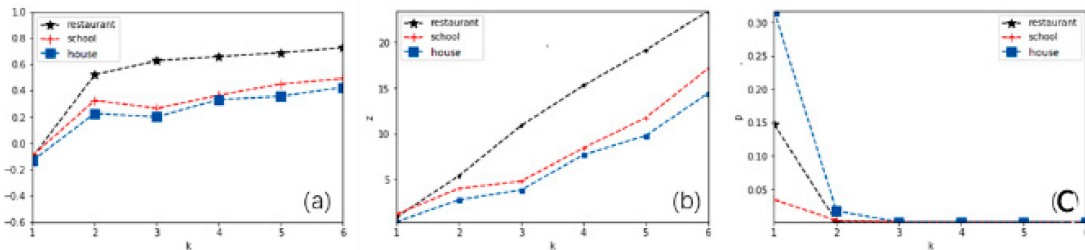

**Figure 14.** Trends of (**a**) *I*, (**b**) *z*, and (**c**) *p* values for points A, B, and C by *vn(k)*.

3.2.2. Variant Phenomenon Detection

As Figure 15 shows that because the vector components of the house type of geo-text were distributed between (72, 90), geo-texts of the house type formed a high-value aggregation region at P7 from the Voronoi 1st-order neighborhood to the Voronoi 3rd-order neighborhood. When *k* was between 4 and 6, inclusive, other types of geo-texts were added into the neighborhood and the vector components of the house type were distributed between (0, 90), and Moran's I could be calculated normally. Therefore, *vn(3)* was a high-value aggregation distribution region of P7, and other types of geo-texts were excluded. In the second type of variation, as shown in Figure 16, the low value is generally between 0 and 12, significantly lower than 20. The Voronoi 1st-order to the Voronoi 3rd-order neighborhoods obey low-value aggregation distribution, and therefore P7 should exclude restaurant types of geo-texts.

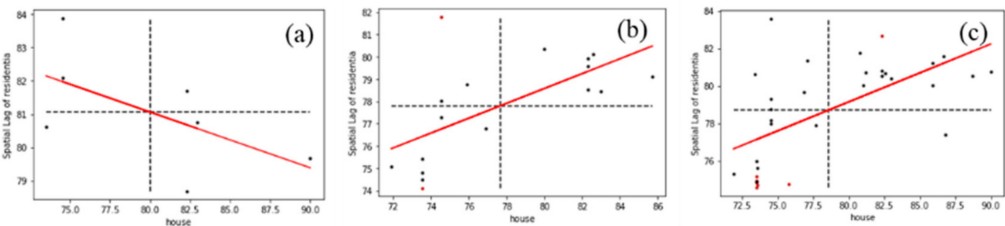

**Figure 15.** High-value aggregation distribution in (**a**) *vn(1)*, (**b**) *vn(2)*, and (**c**) *vn(3)* of P7.

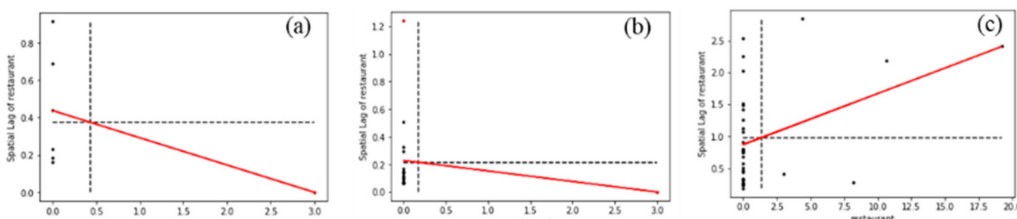

**Figure 16.** Low value aggregation distribution in (**a**) *vn(1)*, (**b**) *vn(2)*, and (**c**) *vn(3)* of P7.

*3.3. Experimental Results Analysis*

3.3.1. Geo-Text Classification Results

The similarity between the VSCR and the geo-text to be input determines whether the annotation to be added is correlated with the neighborhood. Thus, the components of a classification vector are the probability of each geo-text type. In fact, when geo-text vector, like *vc* (0.5,0.5,0,0), provide restaurant and hotel two services, it is hard to say which type a belong to. However, according to the method in this paper, a similar geo-text can be found out.

The field investigation on the classification of 631 records in the experimental area found that seven annotation types were ambiguous to belong to a type and nine were misclassified. The accuracy of the classification was 97.49%. After making a careful analysis of those geo-text, the short length of geotext and a geo-text with serval type information were the important factors result in an error.

### 3.3.2. Result of CGD Algorithm

Geo-text A was placed into 20 random positions, shown in Figure 11, to verify the CGD algorithm. The *VSCRs* were established at the 20 positions, respectively, for each, a convergent k in *VSCRs* was calculated, as shown in Figure 17a. With $\tau = 0.9$ as the minimum convergence k value, the geo-text number in each *VSCR*, correlated geo-texts in the *VSCR*, and the ratio between correlated and uncorrelated geo-texts in the *VSCR* are shown in Tables 2 and 3.

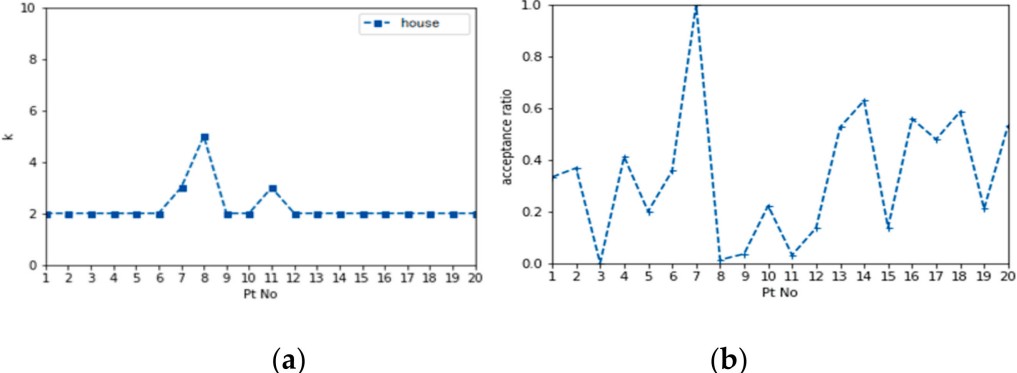

| (a) | (b) |

**Figure 17.** (**a**) Convergence k value for A. (**b**) Correlated-to-uncorrelated ratio of geo-text A.

**Table 2.** Results of the geo-text A test in 20 positions (Pt1-Pt10).

| Pt ID | 1 | 2 | 3 | 4 | 5 | 6 | 7 | 8 | 9 | 10 |
|---|---|---|---|---|---|---|---|---|---|---|
| K | 2 | 2 | 2 | 2 | 2 | 2 | 3 | 5 | 2 | 2 |
| Similar GTAs | 7 | 7 | 0 | 7 | 6 | 10 | 33 | 1 | 1 | 4 |
| DGACR' GTAs | 21 | 19 | 21 | 17 | 30 | 28 | 33 | 78 | 28 | 18 |
| Ratio | 0.33 | 0.37 | 0 | 0.41 | 0.2 | 0.36 | 1 | 0.01 | 0.04 | 0.22 |
| Correlated or not | Y | Y | N | Y | Y | Y | Y | Y | Y | Y |

**Table 3.** Results of the geo-text A test in 20 positions (Pt11-Pt20).

| Pt ID | 11 | 12 | 13 | 14 | 15 | 16 | 17 | 18 | 19 | 20 |
|---|---|---|---|---|---|---|---|---|---|---|
| K | 3 | 2 | 2 | 2 | 2 | 2 | 2 | 2 | 2 | 2 |
| Similar GTAs | 1 | 3 | 10 | 17 | 4 | 19 | 23 | 17 | 4 | 16 |
| DGACR' GTAs | 32 | 22 | 19 | 27 | 29 | 34 | 48 | 29 | 19 | 30 |
| Ratio | 0.03 | 0.14 | 0.53 | 0.63 | 0.14 | 0.56 | 0.48 | 0.59 | 0.21 | 0.53 |
| Correlated or not | Y | Y | Y | Y | Y | Y | Y | Y | Y | Y |

Figure 17b shows the ratio of correlated to uncorrelated geo-texts for each *VSCR*, and the correlation degree established via the CGD algorithm is shown in Figure 18, where larger circles indicate a greater probability that the geo-text was allowed. When a *VSCR* lacked a correlated geo-text, the geo-text was refused from the map, for instance, P3 point. We also could filter uncorrelated geo-text by setting a ratio threshold.

We performed the same procedure with geo-texts B and C and obtained similar ratios, as shown in Figure 19. From these results, we drew several conclusions: P3 was only related to restaurant type, and excluded other geo-text types; P7 was only related to housing type, and other types were filtered; P5, P9 and P12 were densely filled with schools, with scattered restaurants and houses; P8 and P11 were densely filled with restaurants, with other types scattered nearby. At other locations, the distributions appear to be more heterogeneous.

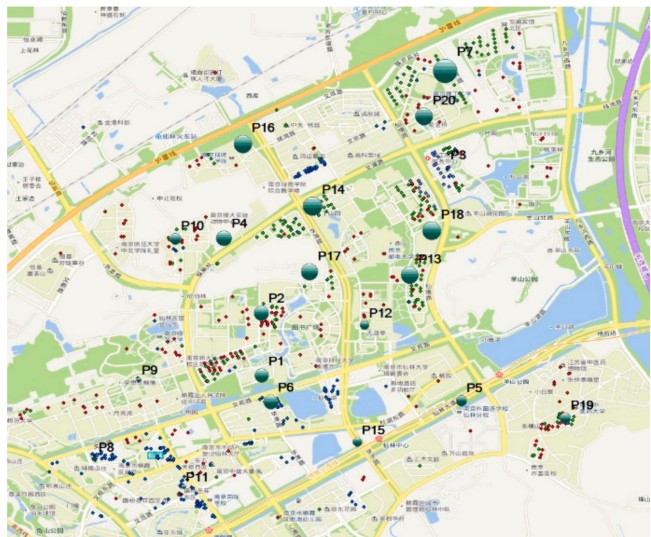

**Figure 18.** Result of the CGD algorithm, where the sizes of the circles represent the correlation degree.

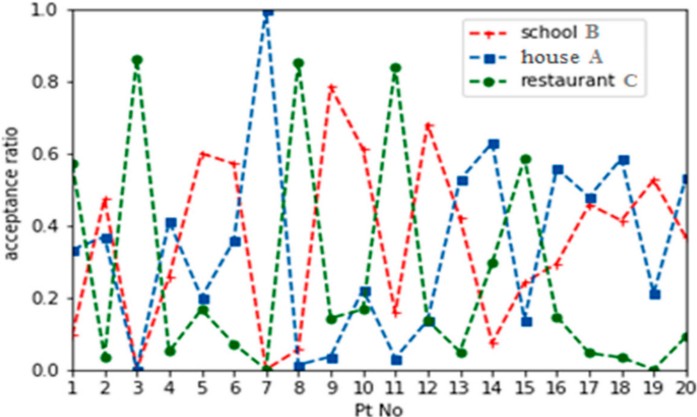

**Figure 19.** Correlated-to-uncorrelated ratios of geo-texts A, B and C.

Through field observations, these descriptions of the 20 areas in the figure were consistent with the actual situation, demonstrating the applicability of the proposed method. Testing in a 64-bit Win10 system environment with 8 GB of memory, the time consumption of a single geo-text analysis was 0.136 seconds on average, which can thus serve actual applications with high efficiency.

## 4. Discussion

As the length of a geo-text is generally short, an appropriate and accurate classifier can be constructed based on TF-IDF characteristics and a polynomial naive Bayes classification model. However, text auto-classification is bound to produce errors, mainly from two sources: (1) an incomplete corpus of samples (e.g., if "KFC" is not included in the corpus when a geo-text appears with the keyword "KFC", that geo-text cannot be well-classified), and (2) fuzzy geo-texts that cannot be easily classified based on semantics or word frequency. For example, one geo-text related to food reads: "There is a parking lot under the ground of the restaurant, the building is very new, the overall satisfaction is very good, worthy to recommendation". Classification results showed that the type had similar probabilities of being residence- or restaurant-related, and thus the semantic distribution of geo-texts in the neighborhood was seriously compromised.

Although we can construct a more comprehensive and accurate corpus to reduce the first type of error, the second type of error is inherent to the uncorrelated description of the geo-text. We, therefore, need to further restrict the length, content, and form of geo-texts to describe features in the map

accurately. However, this goal is difficult to achieve in an open map service. Too many classifications increase the complexity of autocorrelation convergence due to fuzziness, and too few classifications are not enough to describe the distribution of geospatial space. In order to reduce errors, we suggest limiting geo-text categories to five, beyond which more stringent quality, length, and style requirements must be implemented.

Besides, as English text can be segmented with space and punctuation marks, the English geo-text classification process is simpler than the Chinese. Therefore, the idea of the geo-text classification method proposal in this paper is also effective for English geo-text.

The Voronoi natural proximity and the specific weighting method enhance the ability of convergent neighborhood selection. Voronoi k-order selection method eliminates the spatial differences among irregularly distributed data points and uneven densities. The different weighting methods of Euclidean distance and adjacent distance to the location are considered comprehensively to differentiate the influence of the same distance on the different order. The combination of the two methods enables us to further complete the follow-up work of the detection.

The experimental results prove that the method can identify the uncorrelated text which does not conform to the annotated position. However, the method still has some shortcomings. Firstly, we need to know the prior knowledge of geo-text annotation distribution in the research area, a given type range. Only in this way can we select the crawled corpus content to train the classifier, so as to ensure the accuracy of classification. Secondly, the algorithm in this paper provides a spatial correlation suggestion for whether to add annotations or not, but in practical engineering, it is a more complex process to reject an annotation addition. To some extent, it reduces the occurrence of events only from the perspective of spatial correlation. It is a comprehensive process to allow new geo-text to be uploaded by referring to the suggestions provided by the method in this paper and combining with the actual existence of new or changed geometry objects. In addition, it is an effective way to maintain the geo-text annotations semantic environment.

## 5. Conclusions

In order to prevent uncorrelated geo-texts from being uploaded to web map services, this paper investigated a geo-text automatic classification method and the semantic correlation between new and nearby geo-texts. Based on TF-IDF features and a polynomial Bayesian model, a corpus and classifier suitable for geo-texts were constructed. Based on Voronoi k-order neighborhood partition and a minimum neighborhood convergence of spatial autocorrelation, a correlated geo-text detection (CGD) algorithm was implemented. The efficiency and accuracy of this method were demonstrated in an experiment. For given annotation types in a study region, the Voronoi k-order neighborhood semantic convergent region (*VSCR*) of a geo-text could be established for any position. The CGD algorithm effectively detected uncorrelated geo-texts and provided an effective suggestion for preventing them from uploading to the web map environment.

However, the proposed method is not suitable for regions with insufficient density to enable semantic autocorrelation. Many an unattractive building owning few geographic text annotations (or empty) result in decreasing space semantic information density. Some structured data carry a lot of available spatial semantic knowledge, for instance, BIM (building information modeling), CIM (city information modeling), land use data and so on. Hence, future work can add structured data, such as building footprints, land use, and traffic facilities, BIM, CIM, to enrich the semantic descriptions. Furthermore, as our method assumes that the geo-text points have identical connectivity, we did not consider the effect of roads, water systems, and natural features on the segmentation. The spatial division and weighting method considering obstacle limitations should be further improved.

**Author Contributions:** Conceptualization, Yufeng He and Yehua Sheng; Methodology, Yufeng He; Validation, Yufeng He; Formal Analysis, Yufeng He; Investigation, Yufeng He; Resources, Yufeng He; Data Curation, Yue Yin and Yunqing Jing; Writing—Original Draft Preparation, Yufeng He; Writing—Review and Editing, Yehua Sheng

and Ahmad Hasnain; Visualization, Yufeng He; Supervision, Yehua Sheng; Funding Acquisition, Yehua Sheng. All authors have read and agreed to the published version of the manuscript.

**Funding:** This research was funded by the Key Fund of National Natural Science Foundation of China, grant number 41631175; the National Key Research and Development Program of China, grant number 2017YFB0503500.

**Acknowledgments:** The authors express their gratitude towards the journal editors and the reviewers, whose thoughtful suggestions played a significant role in improving the quality of this paper.

**Conflicts of Interest:** The authors declare no conflict of interest.

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
