# Peer review of "Uncorrelated Geo-Text Inhibition Method Based on Voronoi K-Order and Spatial Correlations in Web Maps"

_ijgi, doi:10.3390/ijgi9060381_

Round 1

Reviewer 1 Report

Dear authors,

The article discusses the issue "An uncorrelated method of inhibiting geotext based on the order of Voronoi k and spatial correlations on online maps." The paper presents good problem analysis and reviews possible methods to solve them. In my opinion, the methodology of the solution is very good, and the results are well documented and justified. All chapters are well described in a formal but understandable approach. It is clear that the research method and results are scientifically presented.

Congratulations to the authors of their very interesting research

Here are some small fixes and suggestions:

In the pdf version of the document I received figures that are not of the required quality, so this should be corrected. Details in some drawings are very weak and unclear (especially in Fig. 10 d) e) f)).
Figure No. 9b) should be enlarged so that the reader can orientate himself in the spatial distribution of study objects. There must be a separate drawing, or even two separate drawings so that the study area is visible Geo-text points are visible along with services such as schools, residential buildings or restaurants (e.g. buildings colored to distinguish their functions).
Whether annotations are entered at or outside of the place. Are there buildings like schools or restaurants that don't have any Geo-text? This is te question also. 

Reviewer 2 Report

I like the approach and methods. I think there are some issues to consider in the use of the Pearson correlation statistic and possibly k-clustering approaches could be used to assess the spatial quality of the approach’s results, but more to that later. 

The fundmental problem for the article in my reading is that you don’t explicitly define what geo-text is in terms of its structure and in term of its differences to ‘traditional annotations”. Without clear concepts, any statistical results, regardless how inspiring from a computational point of view, have very limited scientific merit as the semantics are perhaps arbitrary and certainly unverifiable. Hence, I could not positively rate your submission. The Chinese National Language Commission Corpus and International Standard Industrial Classification of All Economic Activities discussion is too abbreviated and much too late. Moving this forward would address my critique. 

In terms of the Pearson correlation the problem is that it assumes a linear relationship and also assumes the data is normally distributed. That is definitely not the case with geographic data. Therefore, I think you need to explain how you deal with this limit or chose a different measure of correlation, Perhaps Kendall Tau? A cross-check of the filtering results with k-means clustering could assess the spatial coherency of the output. 

Other reviewers will likely offer good insights into the statistical approach you develop. 

Reviewer 3 Report

This paper presents a method to investigates the meanings of unstructured geo-text annotations created by voluntary users and assess the degree of spatial autocorrelation in convergence region. The use of the Voronoi-k-order neighborhood self-growth strategy as well as statistical regression analysis, such as the Person method, is very well articulated and clearly explain. In particular, the minimum Voronoi k-order neighborhoods that delineates the space of the geo-text semantic autocorrelation, in other words, Voronoi-k-order semantic convergence region (VSCR) and the precedent creation of an appropriate classifier of an unstructured geo-text based on a term frequency-inverse document frequency (TF-IDF) demonstrates a promising method to filter irrelevant geo-texts that might be included in the web map. Overall, the paper clearly responds to the question of whether or not we can extract means from unstructured annotations, if it is possible to have a meaningful quantitative measure of geographical patterns and correlations, and then also if we can relate such to existing geographical contexts (considering the semantics in convergence region). A comprehensive review of related literature and methods are reviewed and thoughtful compared and considered. I believe a current manuscript contributes greatly to the concept and practice of data-driven geospatial semantics, geographic analysis of natural language processing (NLS), and Machine Learning and GIS.

However, I have one minor point to make. I understand the paper displays an ‘experimental’ case (e.g. experiment validation); however, it is not clear how the authors came up with three geo-texts-A, B, and C-of unknown type and I believe it needs to be explained before it is applied to verify the process of geo-text classification (p.11). As the authors crawled randomly Chinese geo-text instances for each geo-text type from the BaiduMap AP, I wonder it would be better to use randomly chosen geo-texts rather than the ones created by the authors (again, even though we know that it is an experiment). I also believe the original Chinese geo-text before the translation to English is also provided, and the discussion about the role of language in geo-text classification process might be worth to be added.

I also found a typo in the second last sentence on Page 10: “…we can been see from the website…” (Here, “been” should be deleted)
